# Influenza Vaccines toward Universality through Nanoplatforms and Given by Microneedle Patches

**DOI:** 10.3390/v12111212

**Published:** 2020-10-24

**Authors:** Sijia Tang, Wandi Zhu, Bao-Zhong Wang

**Affiliations:** Center for Inflammation, Immunity & Infection, Georgia State University, Atlanta, GA 30303, USA; stang13@gsu.edu (S.T.); wzhu3@gsu.edu (W.Z.)

**Keywords:** influenza, universal influenza vaccine, protein nanoparticles, microneedle patch

## Abstract

Influenza is one of the top threats to public health. The best strategy to prevent influenza is vaccination. Because of the antigenic changes in the major surface antigens of influenza viruses, current seasonal influenza vaccines need to be updated every year to match the circulating strains and are suboptimal for protection. Furthermore, seasonal vaccines do not protect against potential influenza pandemics. A universal influenza vaccine will eliminate the threat of both influenza epidemics and pandemics. Due to the massive challenge in realizing influenza vaccine universality, a single vaccine strategy cannot meet the need. A comprehensive approach that integrates advances in immunogen designs, vaccine and adjuvant nanoplatforms, and vaccine delivery and controlled release has the potential to achieve an effective universal influenza vaccine. This review will summarize the advances in the research and development of an affordable universal influenza vaccine.

## 1. Introduction

The influenza virus belongs to the *Orthomyxoviridae* family [1]. Of the family, both type A and type B influenza viruses are pathogenic to humans [2]. Both type A and type B viruses can cause epidemics, manifested by high death and hospitalization numbers during the yearly flu season [3]. Occasionally, type A viruses can result in influenza pandemics when a new strain containing drifted or shifted antigen acquires the capacity to spread efficiently in humans [4].

Seasonal influenza vaccines are available to prevent epidemics, but the vaccine efficacy is suboptimal because of the rapid accumulation of mutations in circulating strains [5]. A vaccine for influenza pandemics has not been developed. Recent progress in relevant techniques has laid a foundation for developing an influenza vaccine that will induce broad cross-protection to combat influenza epidemics and pandemics [6,7]. This ambitious objective can only be achieved by combining multiple new techniques developed in different aspects of vaccinology, including structure-based immunogen design, optimized vaccine/adjuvant nanoplatforms, and shelf-stable, self-applicable vaccine delivery and controlled release technology.

We will review the progress in immunogen designs, nano technique-based vaccine platforms, and microneedle patch-based skin administration for universal influenza vaccines. We will discuss how a comprehensive universal influenza vaccine approach will integrate all these advances into the future universal influenza vaccine Research and Development (R&D).

## 2. Antigenic Structures Conserved over Different Influenza Types Are Ideal Immunogens for a Universal Influenza Vaccine

Researchers have paid particular attention to conserved influenza immunogens, especially the conserved structures in influenza surface antigenic proteins [8]. Both type A and type B influenza viruses contain the major surfaces antigens, hemagglutinin (HA) and neuraminidase (NA), which can be categorized into 18 HA subtypes (serotypes) across two phylogenic groups and 11 NA subtypes for type A viruses [9]. Figure 1 displays the influenza virus diagram with different structured and unstructured proteins for vaccine antigens.

### 2.1. Hemagglutinin Stalk Domain

HA is an essential protein for viral pathogenesis and antigenicity. Although HA is highly mutable, some structural features are conserved between phylogenic groups or subtypes [10]. A monoclonal antibody (mAb) was found to recognize a very conserved sequence (Arg 118, Asp 151, Arg 152, Arg 224, Glu 276, Arg 292, Arg 371, Tyr 406) in the membrane-proximal stalk domains of both type A and type B influenza [11]; this mAb was broadly protective and broadly neutralizing. Broadly neutralizing Abs (bnAbs) recognizing the conserved HA stalk domains of an individual group demonstrated shared antigenic structures at the phylogenic group level [12]. Some such conserved structures have been accurately deciphered [13]. Vaccines using these conserved antigenic determinants can induce broadly reactive immune responses crossing different influenza types, phylogenic groups, or subtypes.

Several research laboratories, including ours, have successfully constructed and tested recombinant proteins retaining the conserved HA stalk structures without the immunodominant HA head domain as universal vaccine immunogens [14,15,16,17]. Wild-type HA are trimers. Foreign trimerization sequences or scaffolds have been used to stabilize the trimerization of these HA stalks [18]. To some extent, these designs have improved the immunogenicity of the conserved structures but mainly induce non-neutralizing antibody responses [14].

### 2.2. Neuraminidase

Neuraminidase (NA) is another essential influenza surface antigen. Compared to HA, NA undergoes much lower antigenic drift and shift and is more suitable for influenza vaccine immunogens in terms of vaccine universality [19]. However, NA is not as immunogenically impactful as HA in seasonal influenza vaccination or influenza infection owing to HA immunodominance [20,21]. Given in a vaccine formulation without the immune shielding effect of other strong immunogens, NA can induce immune responses conferring broader protection. Some NA-specific monoclonal antibodies have been identified from humans recently. Studies have demonstrated that these monoclonal antibodies can therapeutically protect mice from lethal doses of homo- and heterologous influenza infection [22,23]. A universally conserved NA epitope between 222–230 induced NA-inhibiting (NAI) antibodies against all influenza types [24]. The evidence indicates that NA has the potential to be developed into a universal influenza vaccine, or a synergistic component of such vaccines, if it is presented in an immunogenic form without the immunodominance of other antigens, such as protein nanoparticles [25].

### 2.3. The Ectodomain of Matrix Protein 2

Compared to HA and NA, the third membrane protein—influenza matrix protein 2 (M2)—is much smaller and less immunogenic. M2 is a homo-tetrameric ion channel and plays a vital role in uncoating viruses after viral entry. M2 is expressed as an integral transmembrane protein composed of 97 amino acid residues. The 24 amino acid residues at the N-terminus form the conserved ectodomain (M2e) [26,27,28,29]. Although the M2 protein does not induce a robust immune response during the natural viral infection due to its small size and a low count on the virion surface, researchers have increased M2e immunogenicity by designing polymeric peptides of M2e [30], fusing M2e with innate signaling sequences [31], or incorporating M2e into nanoparticle systems to induce high levels of M2-specific immunity [32,33]. M2e-specific antibodies are not neutralizing, but their protective effects were demonstrated in many studies, and M2e-specific T cell response is also critical [34].

Although most M2e vaccine immunization reduced morbidity in laboratory animal studies, immunized animals showed sickness symptoms during live virus challenges. Therefore, M2e-based vaccines may not be sufficient for a standalone universal influenza vaccine but could be a synergistic part of such vaccines. M2e-based vaccines have been combined with seasonal vaccines and immunizations and induced more comprehensive immune protection than the seasonal vaccine alone [35]. Although M2e could be used to protect people from influenza A, it could not be used against influenza B because the M2 protein in influenza B is structurally different [3].

### 2.4. Influenza Nucleoprotein

As a structural protein, influenza nucleoprotein (NP) is the most abundant viral protein in infected cells. Unlike HA and NA, NP is highly conserved [36]. Many reports demonstrated that NP-specific T cell responses, including CD4+ and CD8+ T cell responses, contributed to heterosubtypic protection and long-term immune memory [37,38,39,40,41]. Some conserved HLA I and II-recognized epitopes have been characterized. Fusion peptides and recombinant proteins including these epitopes have been studied in laboratory animals and showed strong T cell responses and broad protection [42,43,44,45]. However, novel approaches are to be investigated to identify and utilize new NP T cell epitopes to design novel immunogens for a universal influenza vaccine. Several studies have already used nanoplatforms to develop universal influenza vaccines with NP [46,47]. However, there are gaps in fully deciphering and mapping NP T cell epitopes and utilizing these sequences toward influenza vaccine universality.

## 3. Nanoplatforms for Conserved Influenza Structures

Because a conserved influenza structure-based vaccine will exclusively induce broadly reactive immune responses, these structures are ideal as immunogens for universal influenza vaccines [6]. An apparent weakness of the conserved influenza determinants is their low immunogenicity. Nanotechnology has shown great potential in generating a broadly protective universal influenza vaccine with the conserved influenza immunogens.

Many features of nano-sized vaccines resemble a viral particle in terms of antigen size, distribution, exposure, retention, and presentation. The human immune system has evolved to battle viral infection for millions of years and can effectively sense, retain, process, and respond to this biomimicry [48,49]. In addition to these physical features that contribute to improved immunogenicity [50], nanoparticulate vaccine platforms can facilitate the realization of universal influenza vaccines by improving several other critical immunogenicity-determining qualities of a vaccine candidate, such as antigen spatial orientation, controlled release, and adjuvant integration.

The orientation of conserved antigenic structures on nanoparticle surfaces can resemble the antigen’s spatial environment in influenza viruses. Ferritin nanocages can correctly orientate the HA stalk by functioning as a scaffold for the HA stalk. The tri-axial arrangement of ferritins in the cage provides the HA stalk antigen the spatial environment to exclusively form trimeric spikes [51]. These spikes showed the conformation of natural HA stalks and triggered antibody responses to conformational determinants. A different approach to displaying conformational determinants is to express the HA stalk domain in trimeric forms and then assemble the trimeric recombinant proteins into nanoparticles [14,52,53,54]. Recently, a layered protein nanoparticle platform has been found to induce a strong antibody response to the HA stalk on the surface of the nanoparticle and T cell responses to the antigens within the nanoparticle core, which resembles the antigenicity of viral surface and internal antigens [14,45].

The controlled-release of antigens can be embedded into the design of a nano-vaccine platform. Chemical and physiological change-derived antigen-release in specific extra- and intracellular compartments guarantee the right immune cell encounters. Our previous reports have shown that DTSSP fixed layered protein nanoparticles disassembled exclusively in intracellular redox conditions, endowing the vaccine proteolysis-resistance in the extracellular space after vaccination and improved antigen-retention [14]. The layered designs of the particulate vaccines can provide a phase-pulsed antigen release pattern, controlling the timing to favor phase-based immune response development [55].

The integration of antigens and adjuvants in a nano-vaccine platform can program a comprehensive immune response. Compared to soluble recombinant protein antigens, nanoparticles have a much larger size. The particle surface has a high capacity for the polymeric orientation of antigens and adjuvants. The interior space of particles can accommodate different antigens and adjuvants to meet the varying needs for antigen-presentation. In one of our studies, we found that gold-nanoparticle surfaces can be differentially modified to conjugate HA and flagellin, a ligand of TLR5. The conjugated flagellin can trigger both cell surface TLR5 innate signaling pathways and cytosolic receptor IPAF-mediated inflammasome pathways, instead of the TLR5-signaling pathway alone by soluble flagellin, resulting in enhanced immune responses to HA conferring heterologous protection in mice [56,57].

Although individual approaches have improved the immunogenicity of these conserved influenza structures, a comprehensive nano vaccine platform integrating most of these advanced features is to be investigated. Furthermore, other drug delivery techniques may be combined with the nano vaccine platforms in alternative vaccination routes to induce complementary immune responses. Our protein nanoparticle designs and immunization strategies are shown in Figure 2.

## 4. Different Route for Vaccine Delivery

Different routes had been applied for vaccine delivery. Generally, vaccination routes include intramuscular injection, skin delivery, intranasal and oral administration [58]. Intramuscular injection is widely used for immunization because it is easy to apply with a low risk of adverse reactions [59]. However, the pre-injection anxiety and post-injection pain caused by intramuscular injection cannot be avoided [60]. Vaccine delivery by oral administration is challenged by mucosal barriers and degradation by a harsh gastrointestinal (GI) environment [61]. Some studies showed that intranasally vaccination could also induce strong and sustained T-cell responses [62]. The oral and intranasal routes are more efficient for infectious diseases because most infections are initiated at the mucosal surface [63]. Intranasal administration could be a promising method to induce a strong mucosal immune response to fight against influenza viral infection. The skin has been recognized as the largest immune organ and contains abundant lymphatic vessels and various immune cells in the dermis. Skin resident lymphocytes, including Langerhans cells, dermal DCs, and macrophages, are important for the induction of innate immune response and development of adaptive immune response. A novel, thin-film technology had been developed to deliver the vaccine through buccal, and it could at least induce comparable antibody responses as intramuscular injection [64]. Comparing to the intramuscular route, vaccination through the skin has been demonstrated to induce improved immune responses and provide better protection [65,66].

## 5. Dry Formulation of Influenza Vaccines on Microneedle Patches

With the development of novel vaccine-delivery techniques, painless, simple-to-administer microneedles have also been used for influenza vaccine delivery and enhanced immune protection. Microneedle arrays can penetrate the stratum corneum, the outer layer of the skin. The vaccines are encapsulated into tiny sharp tips. By skin administration, vaccines or drugs will dissolve into the epidermis and dermis [67,68,69]. Many different immune cells in the skin, including Langerhans cells, αβ T lymphocytes and dendritic epidermal T cells (DETCs) in the epidermis, dDCs, macrophages, eosinophils, neutrophils, mast cells, γδ T lymphocytes and B lymphocytes in the dermis, and T as well as B cells in the skin-draining lymph nodes, indicate that the skin is an attractive site for vaccination [70]. Furthermore, the simplified administration of a universal influenza vaccine would significantly reduce the morbidity and mortality from a newly emerging influenza pandemic when resources such as vaccine production, storage, transportation, and healthcare service facilities are limited [71].

Combining nano universal influenza vaccines with the microneedle patch vaccination route could meet the extraordinary need for the rapid distribution of an effective vaccine worldwide upon an influenza pandemic emergency. In line with this need, microneedles encapsulating the fusion protein 4.M2e-tFliC or virus-like nanoparticles have been studied [71,72,73]. In our studies, soluble protein or nanoparticles encapsulated MNP delivered antigens into the epidermis and dermis of the skin. Compared to intramuscular injection, MNP immunization induced promoted DCs migration to the draining lymph nodes and enhanced germinal center (GC) reactions, including elevated GC B cells and follicular helper T (Tfh) cells [45,74]. The dose-sparing effect of microneedle delivery provides an excellent benefit for preventing an emerging influenza pandemic because the available vaccine production capacity can yield more vaccine doses. Our recent data have shown that dissolvable microneedle patches incorporating NA/M2e layered protein nanoparticles induce NA antibody responses with much higher NA-inhibition titers [75]. With the advantages of the microneedle patch vaccination route, skin vaccination delivering integrated nano vaccines is a promising approach for moving influenza vaccines toward universality.

## 6. Conclusions

Universal influenza vaccines are feasible, although gaps are still to be filled to the successful R&D of such vaccines. Challenges include identifying conserved epitopes, improving the immunogenicity of weakly immunogenic conserved epitopes, efficiently fabricating nano-vaccine candidates with conserved epitopes, formulating nano-vaccines into stable formats such as dissolvable microneedles, and inducing broadly protective immune responses. Time is of the essence because the next influenza pandemic is not a question of ‘if’ but ‘when’.

## Figures and Tables

**Figure 1 viruses-12-01212-f001:**
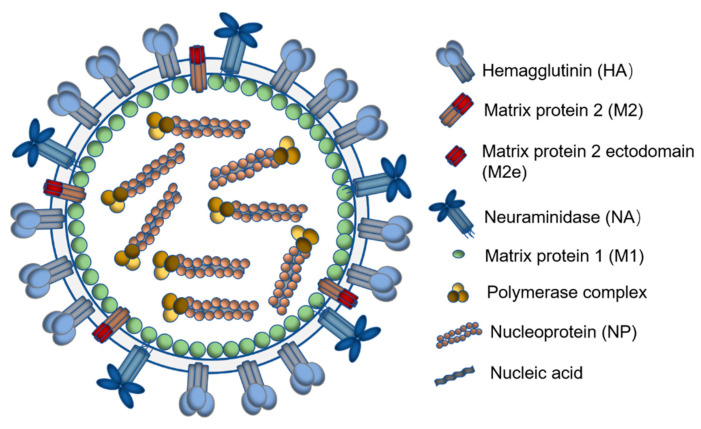
Schematic diagram of the influenza virus showing antigenic viral proteins.

**Figure 2 viruses-12-01212-f002:**
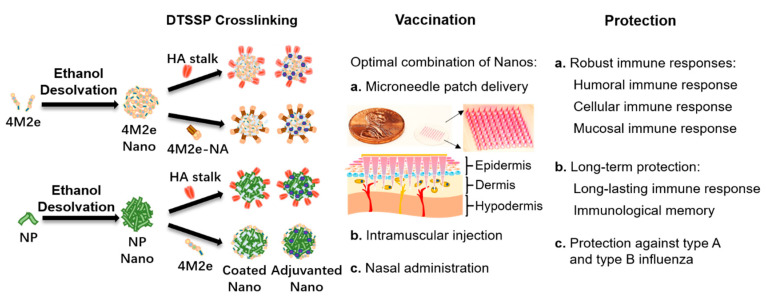
Diagram of nanoparticle creation from conserved immunogens, vaccination routes, and protection. At left: A diagram illustrating the formation of layered protein nanoparticles through ethanol desolvation of immunogens into core particles followed by DTSSP crosslinking of an outer layer of immunogens onto the core. At center: A list of possible vaccination routes with protein nanoparticles. At right: A list of protective immune response goals for a universal influenza vaccine.

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
