# Peer review of "Influenza Vaccines toward Universality through Nanoplatforms and Given by Microneedle Patches"

_viruses, 2020, doi:10.3390/v12111212_

Round 1

Reviewer 1 Report

This review deepens the knowledge on the recent advances in the development of a universal influenza vaccine. The work is interesting and informative. However, I have some observations that are listed in the following lines.

  1. Line 23: Both Type A and Type B…
  2. Line 25: “… when a new strain acquires the capacity for transmission in humans”, Explain.
  3. Line 29: Provide a reference.
  4. Line 31: This ambitious…
  5. Line 38: …Research and Development (R&D).
  6. Lines 41-42: Rephrase.
  7. Lines 46 and 62: Avoid using abbreviations as a subtitle.
  8. Line 49: “… a very conserved sequence”, Add the sequence and show how it is conserved using a multiple sequence 
  9. Line 50: … this mAb…
  10. Line 108: Provide a reference.
  11. Line 152: The diagram is too small to see the details.
  12. Line 191: grammar.
  13. Add a list of abbreviations.

Author Response

Reviewer 1:

This review deepens the knowledge of the recent advances in the development of a universal influenza vaccine. The work is interesting and informative. However, I have some observations that are listed in the following lines.

Q1: Line 23: Both Type A and Type B…

A1: Change has been made, as suggested (highlighted in lines 22-23).

Q2: Line 25: “… when a new strain acquires the capacity for transmission in humans”, Explain.

A2: We explain this sentence more clearly (highlighted in lines 25-26).

Q3: Line 29: Provide a reference.

A3: References have been added (highlighted inline 31).

Q4: Line 31: This ambitious…

A4: Changes have been made (highlighted in line 31).

Q5: Line 38: …Research and Development (R&D).

A5: Changes have been made (highlighted in line 38).

Q6: Lines 41-42: Rephrase.

A6: Changes have been made (highlighted in lines 41-42).

Q7: Lines 46 and 62: Avoid using abbreviations as a subtitle.

A7: Changes have been made (highlighted in lines 51 and 68)

Q8: Line 49: “… a very conserved sequence”, Add the sequence and show how it is conserved using a multiple sequence 

A8: Changes have been made (highlighted in lines 54-55)

Q9: Line 50: … this mAb…

A9: A change has been made (highlighted in line 56)

Q10: Line 108: Provide a reference.

A10: A reference has been added (highlighted in line 114).

Q11: Line 152: The diagram is too small to see the details.

A11: The diagram is enlarged.

Q12: Line 191: grammar.

A12: We have removed this section according to Reviewer 3’s comment.

Q13: Add a list of abbreviations.

A13: Abbreviations are carefully added behind the initial words or phrases when they appear for the first time.

Reviewer 2 Report

The manuscript by Tang et al is a highly biased review of the potential for a universal vaccine based on conventional influenza antigens and novel delivery methods, particularly nanopatches. It is wholly non-original as each of these topics is well covered in the current literature (“Towards a universal flu vaccine” was covered by Nature in Sept last year and microneedle reviews for drug and vaccine delivery are everywhere) and it ignores other routes to vaccination. Vectored vaccines for example, are not discussed, nor are other methods of delivery, for example under the tongue films. The title is misleading as it implies that a universal flu vaccine is imminent, as does the rather grandiose statement of “if not when” in the final line. But it fails to note the failure in trial of the “Palese” vaccine last year which caused GSK to “dump” it as a project. And although the M2e protein is mentioned the text fails to stress that it does not induce neutralizing antibodies and is thought to work, if at all, through opsonization. As with the GSK stalk vaccines, to my knowledge Sanofi are not actively pursuing it as a viable vaccine.

Strangely, the only figure presented for such a wide topic is of vaccine designs which are very, very far from the clinical as they are so complicated to make their supply will always be limited, another example of the massive bias in what is being covered here.  

The English of the article is acceptable but not good. The problem is imprecision throughout. For example we have “T-epitopes” instead of T-cell epitopes (I guess T epitopes could be those containing threonine) and “conserved influenza immunogens have ….conserved structures” - a bit obvious perhaps. Extensive editing of English language is required.

At the end we leave influenza altogether and list other microneedle demonstration vaccines (none in clinical use). The section is clearly an afterthought and does not fit at all. The review states influenza and it should restrict itself to that. You could add ten such sections on other vaccines, not least, today, Covid, but what’s the point?

Author Response

Reviewer 2:

Q1: The manuscript by Tang et al. is a highly biased review of the potential for a universal vaccine based on conventional influenza antigens and novel delivery methods, particularly nanopatches. It is wholly non-original. Each of these topics is well covered in the current literature (“Towards a universal flu vaccine” was covered by Nature in Sept last year and microneedle reviews for drug and vaccine delivery are everywhere) and it ignores other routes to vaccination. Vectored vaccines for example, are not discussed, nor are other methods of delivery, for example under the tongue films. The title is misleading as it implies that a universal flu vaccine is imminent, as does the rather grandiose statement of “if not when” in the final line. But it fails to note the failure in trial of the “Palese” vaccine last year which caused GSK to “dump” it as a project. And although the M2e protein is mentioned the text fails to stress that it does not induce neutralizing antibodies and is thought to work, if at all, through opsonization. As with the GSK stalk vaccines, to my knowledge Sanofi are not actively pursuing it as a viable vaccine.

Strangely, the only figure presented for such a wide topic is of vaccine designs which are very, very far from the clinical as they are so complicated to make their supply will always be limited, another example of the massive bias in what is being covered here.  

 A1: Thank you for your comments. We have made some modifications based on your comments. Other vaccination routes have been discussed (highlighted in lines 166-183). M2e specific antibodies have been mentioned (highlighted in lines 91-92). Overall, we discussed nanoplatforms, microneedle patch-based skin delivery, and our perspectives about the future application of these techniques in the development of universal influenza vaccines. We were not to summarize the advancement of universal influenza vaccine research and development.

Q2: The English of the article is acceptable but not good. The problem is imprecision throughout. For example we have “T-epitopes” instead of T-cell epitopes (I guess T epitopes could be those containing threonine) and “conserved influenza immunogens have ….conserved structures” - a bit obvious perhaps. Extensive editing of English language is required.

 A2: Thank you for your suggestions. The language is further edited.

Q3: At the end we leave influenza altogether and list other microneedle demonstration vaccines (none in clinical use). The section is clearly an afterthought and does not fit at all. The review states influenza and it should restrict itself to that. You could add ten such sections on other vaccines, not least, today, Covid, but what’s the point?

A3: We agreed with the comments but this paragraph was added according to the editor’s suggestion when we submitted this manuscript. Now we removed it.

Reviewer 3 Report

Tang et al. review about influenza vaccines using microneedle patches. Some lack of information should be added before publication described in comments for authors.

Major comments:

  1. A figure depicted the (domain) structures of antigenic proteins (HA, NA, M2, and nucleoprotein) would help readers understand. Please consider adding the figure somewhere.
  2. An explanation of how microneedle patches containing antigenic proteins induce host immune responses is entirely missing. Please add the sentences and a figure.
  3. A comparison of microneedle with other delivery methods is entirely missing. Please address this somewhere.

Minor points:

  1. Line 23: Please add "and" between type and A.

Line 96: Please replace "CD4 and CD8 responses" with "CD4+ and CD8+ T cell responses".

Author Response

Reviewer 3:

Tang et al. review about influenza vaccines using microneedle patches. Some lack of information should be added before publication described in comments for authors.

Major comments:

Q1. A figure depicted the (domain) structures of antigenic proteins (HA, NA, M2, and nucleoprotein) would help readers understand. Please consider adding the figure somewhere.

A1: We have added a figure to show different antigenic proteins, as suggested (highlighted in Figure 1, lines 45-46).

Q2. An explanation of how microneedle patches containing antigenic proteins induce host immune responses is entirely missing. Please add the sentences and a figure.

A2: We have explained the MNP induced immune responses that we found in our studies (highlighted in lines 200-204).

Q3. A comparison of microneedle with other delivery methods is entirely missing. Please address this somewhere.

A3: A paragraph has been added to introduce different immunization routes (highlighted in lines 166-183)

Minor points:

Q1. Line 23: Please add "and" between type and A.

A1: It had been modified (highlighted in lines 22-23).

Q2. Line 96: Please replace "CD4 and CD8 responses" with "CD4+ and CD8+ T cell responses".

A2: They have been replaced (highlighted in line 103).